# A Simple Mean-Teacher UNet Model for Efficient Abdominal Organ Segmentation

Zixiao Zhao[0000−0003−2808−4659] and Jiahua Chu[0000−0001−5031−3046]

AI Innovation and Commercialisation Centre, NUSRI, Suzhou, China
{zixiao.zhao, jiahua.chu}@nusri.cn

**Abstract.** One inevitable barrier to deep learning-based medical image segmentation algorithms is that for such tasks requiring high accuracy, all models must be trained using large datasets annotated by experts, and this process is exceptionally time-consuming and laborious. For abdominal organ segmentation, this problem becomes more prominent as the image size becomes larger. To address this problem, we design a classical UNet model using the Mean-Teacher strategy to obtain relatively satisfactory segmentation (58.93% DSC and 59.54% NSD)results on a semi-supervised abdominal segmentation dataset. The core idea is to use labeled data to improve the segmentation performance of the model itself, while introducing noise on unlabeled data to improve the generalization of the model. Inspired by nnUNet, we use as simple a model structure as possible, thus ensuring the efficiency during training and inference phases ($< 2$GB VRAM consumption and $\sim$10s inference time).

**Keywords:** Medical Image Segmentation · Abdominal Organ Segmentation · Semi-supervised · UNet · Mean-Teacher

## 1 Introduction

In recent years, Convolutional Neural Networks (CNNs) and Transformers-based approaches have achieved state-of-the-art results in the field of medical image segmentation, e.g. [19,1]. However, with the development of such methods, the structure of the model becomes more and more complex, the parameters of the model increase dramatically, and the size of the annotated data required to train such complex models becomes larger and larger [8]. For medical image segmentation tasks, the annotation of the dataset implies expert labeling at pixel or voxel level, a process that is often extremely time-consuming and laborious [18]. For abdominal organ segmentation, this problem becomes more serious because the organs or diseases contained in this region are more complex, and the size and resolution of the images become larger [10].

In this context, semi-supervised segmentation methods become more practical due to their properties of requiring only a small amount of fine annotation and more unlabeled data instead. In the last three years, a large number of semi-supervised segmentation methods have achieved satisfactory results in their respective domains. One of the most widely used methods is the Mean-Teacher

model [15] and its many variants [12,18,17]. Other commonly used strategies include pseudo labeling [16], adversarial learning [6], contrastive learning [11] and etc.

Despite advances in semi-supervised learning benchmarks, previous methods still face several major challenges: **Domain variation:** Most of these methods are based on 2D natural images and require additional learning costs if migrated to medical images. **Generalization:** Considering the limited amount of training data, training deep models is usually deficient due to over-fitting and co-adapting [17].

In this work, we propose a simple and effective semi-supervised scheme that is also based on the Mean Teacher [15] idea. This framework takes labeled and unlabeled images as input and introduces random noise for contamination, respectively. The uncontaminated original input images will predict the results by a Student model composed of an ordinary UNet [13], while the contaminated data will predict the other set of results by a Teacher model with exactly the same structure. For the labeled data, the Student model is supervised by ground truth on the one hand and by the consistency constraint of the predicted results of the contaminated data on the other hand, while for the unlabeled data, only their consistency loss is used for supervision. The parameters of the Teacher model are then periodically updated from the $M_S$ by exponential moving average (EMA).

The main contribution of this work are two-fold: 1) Inspired by nnUnet [7], our approach uses only the classical UNet model for segmentation, making the training and prediction process cheap ($<$5GB RAM and $<$2GB VRAM) and efficient (6s/image). 2) Still inspired by nnUnet [7], we use proper preprocessing methods (and multiple augmentation methods during training phase), which enables our model to achieve stable results even on data with inconsistent distribution.

## 2    Method

### 2.1    Preprocessing

Thanks to the rich transformation API provided by MONAI framework [3], we applied many pre-processing methods that can increase the reusability of the model.

**General preprocessing**: General preprocessing represents transforms that are applied in the training, validation and prediction phases.

- Orientation matching: Based on the orientation of training data, all input images are uniformly adjusted to the "LPI" orientation.
- Resampling method for anisotropic data: After orientation matching we resample the image to the spacing of (4, 4, 10) to reduce the size of the input data.
- Intensity normalization method: For the intensity of the data, we only reserve the voxels whose intensity is inside the interval [-1000, 500], and then adjusted the value range to [0.0, 1.0].

## 2.2   Proposed Method

For general semi-supervised learning, the training set always consists of two parts. The labeled dataset $D_l$ with $N$ annotated images and the unlabeled dataset $D_u$, where there are $M$ raw images ($M >> N$). The whole training set is $D_{N+M} = D_l \cup D_u$. For an image $x_i \in D_l$, its ground truth is available. Conversely, if $x_j \in D_u$, its ground truth is not provided [9]. Our Mean Teacher UNet model is shown in Figure 1. For both $D_l$ and $D_u$, they will be used for the calculation of consistency loss, corresponding to $L_{c1}$ and $L_{c2}$ in the figure. For $D_l$, it is additionally used to compute the common supervised segmentation loss $L_s$ to update the model parameters.

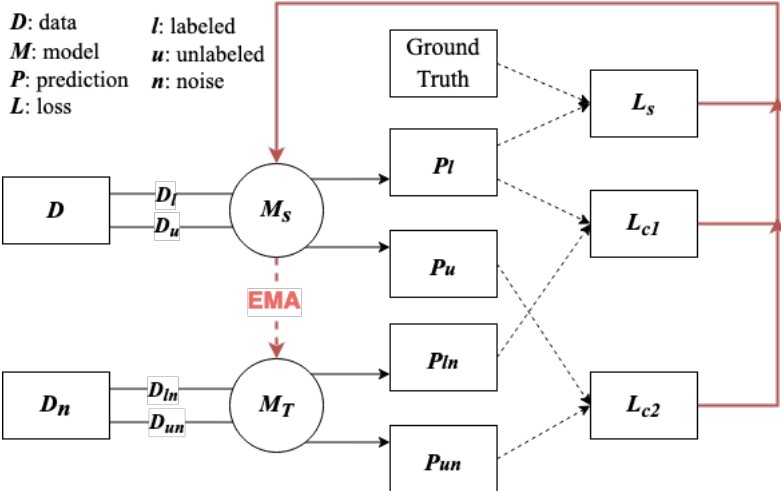

Fig. 1: Network architecture: Student and Teacher model are both randomly initialized, which receive uncontaminated and contaminated data respectively. Teacher model's parameter will be gradually updated from Student model by EMA.

In fact, we followed the exact same strategy as Mean Teacher. The overall architecture of the network consists of two parts, Student model $M_S$ and Teacher model $M_T$. In our design, these two models are composed of two identical initialized UNet models.

$$\theta_T' = \alpha\theta_T + (1 - \alpha)\theta_S \tag{1}$$

The update of $M_T$'s parameters is obtained by exponentially moving average from $M_S$'s parameters, depicted in equation 1. At the beginning of training phase, since model comes from random initialization, the parameters of $M_S$ are definitely incorrect. $M_T$ should be based on what $M_S$ learns, so $\alpha$ should start

from zero. As the network is being trained, after $M_S$ reaches a certain accuracy, the ensemble can eventually be used, which means $\alpha$ can come to the value of 0.99 in the end. The network parameters of the $M_S$ are updated by the gradient descent of the loss function. The loss function includes two categories: first the supervised Dice loss, which ensures the model has the basic segmentation ability, the second part is the unsupervised loss function, or consistency loss, and here we use MSE loss, which mainly ensures that the prediction of $M_S$ is as similar as possible to the one of $M_T$ between the contaminated and uncontaminated data (the contamination applied here is the additive Gaussian white noise). Because the parameters of $M_T$ are the moving average of $M_S$, the prediction should not have too much jitters for any fluctuations. If the model is correct, the predicted labels of the two models Student and Teacher should be close. Then tuning the model in the direction that makes the prediction of the two models close is equal to move the model towards predicting the correct labels.

### 2.3   Post-processing

Due to the nature of the dataset, we did not use specfic post-processing methods.

## 3   Experiments

### 3.1   Dataset and evaluation measures

The FLARE2022 dataset is curated from more than 20 medical groups under the license permission, including MSD [14], KiTS [4,5], AbdomenCT-1K [10], and TCIA [2]. The training set includes 50 labelled CT scans with pancreas disease and 2000 unlabelled CT scans with liver, kidney, spleen, or pancreas diseases. The validation set includes 50 CT scans with liver, kidney, spleen, or pancreas diseases. The testing set includes 200 CT scans where 100 cases has liver, kidney, spleen, or pancreas diseases and the other 100 cases has uterine corpus endometrial, urothelial bladder, stomach, sarcomas, or ovarian diseases. All the CT scans only have image information and the center information is not available.

The evaluation measures consist of two accuracy measures: Dice Similarity Coefficient (DSC) and Normalized Surface Dice (NSD), and three running efficiency measures: running time, area under GPU memory-time curve, and area under CPU utilization-time curve. All measures will be used to compute the ranking. Moreover, the GPU memory consumption has a 2 GB tolerance.

### 3.2   Implementation details

**Environment settings**  The development environments and requirements are presented in Table 1.

Table 1: Development environments and requirements.

| | |
|---|---|
| Windows/Ubuntu version | Ubuntu 18.04.4 LTS |
| CPU | Intel(R) Xeon(R) Gold 6226 CPU @ 2.70GHz |
| RAM | 12×32GB; 2.67MT/s |
| GPU (number and type) | 8× NVIDIA GeForce RTX 2080Ti |
| CUDA version | 11.1 |
| Programming language | Python 3.6.10 |
| Deep learning framework | Pytorch (Torch 1.7.0, torchvision 0.8.0) |
| Specific dependencies | monai 0.8.0 |
| (Optional) Link to code | https://github.com/SeanCho1996/MeanTeacher3dUNet |

**Training protocols** A refined training parameters are shown in Table 2.

In the training phase we perform a series of augmentation on the input data to improve the robustness of the model.

- Random Affine: In this stage we add random rotation and scale transformation.
- Cropping strategy: The cropping strategy is different for labeled and unlabeled training data: for labeled data, the foreground patches are randomly cropped according to the value of the labels, and conversely for unlabeled data, a completely random cropping is used. Patch size is fixed to (128, 128, 16)
- Other augmentation methods: random Gaussian noise as well as random flip in the three axes.

Table 2: Training protocols.

| | |
|---|---|
| Network initialization | "he" normal initialization |
| Batch size | 8 * 3 samples per image |
| Patch size | 128×128×16 |
| Total epochs | 1000 |
| Optimizer | Adam |
| Initial learning rate (lr) | 1e-4 |
| Lr decay schedule | / |
| Training time | 15 hours |
| Number of model parameters | 3.5M[1] |
| Number of flops | 30.27G[2] |
| $CO_2$eq | 1 Kg[3] |

## 4    Results and discussion

### 4.1    Quantitative results on validation set

The overall quantitative results are shown in Table 3.

Table 4 illustrates the results of either using the unlabeled data or not. It can be easily seen that the semi-supervised model outperforms the fully supervised model using only labeled data on all other classes except Pancreas and Duodenum with a subtle advantage of ∼0.6%. The generalization of the model is greatly enhanced due to the use of unlabeled data, coupled with a wide variety of data augmentations.

Table 3: Quantitative results on validation set.

| Organ | DSC(%) | NSD (%) |
|---|---|---|
| Liver | 81.56±17.07 | 72.48±19.02 |
| Right Kidney | 69.03±24.96 | 61.02±24.81 |
| Spleen | 76.03±19.49 | 67.03±20.68 |
| Pancreas | 54.87±14.79 | 65.90±14.47 |
| Aorta | 79.94±12.27 | 76.32±14.21 |
| Inferior Vena Cava | 68.10±14.09 | 58.75±14.45 |
| Right Adrenal Gland | 38.55±17.90 | 51.25±19.69 |
| Left Adrenal Gland | 35.97±20.06 | 47.41±23.77 |
| Gallbladder | 32.81±27.87 | 24.31±21.31 |
| Esophagus | 54.05±15.88 | 65.11±16.99 |
| Stomach | 57.32±19.91 | 53.76±19.39 |
| Duodenum | 46.20±15.99 | 66.54±17.25 |
| Left Kidney | 71.78±22.57 | 64.17±24.43 |
| Mean | 58.93±18.68 | 59.54±19.27 |

### 4.2    Qualitative results on validation set

At the image level, we find that our model performs well in processing test images that are isotropic with labeled data, as shown in Figures 3 and 2. The dimensions of these two images are (512, 512, 96) and (512, 512, 89), respectively, while the average size of the labeled data is approximately (512, 512, 100). Conversely, for images anisotropic with labeled data, as shown in Figures 4 and 5, our model performs relatively poorly in this case. The dimensions of these two images are (512, 512, 203) and (512, 512, 171), respectively, and the scale in the coronal direction is almost twice of the labeled data. The reason for this situation is that in order to reduce the resource consumption of the model, we set the spacing of preprocessing relatively large, and in the process of downsampling, too much information is lost from these large scale images, resulting in their features not being easily computed.

Table 4: DSC(%) comparison on validation set.

| Organ | with unlabeled data | without unlabeled data |
|---|---|---|
| Liver | **85.58** | 80.81 |
| Right Kidney | **71.69** | 67.66 |
| Spleen | **76.07** | 72.87 |
| Pancreas | 53.93 | **54.30** |
| Aorta | **79.62** | 77.67 |
| Inferior Vena Cava | **68.40** | 66.84 |
| Right Adrenal Gland | **38.06** | 37.05 |
| Left Adrenal Gland | **37.91** | 33.03 |
| Gallbladder | **34.22** | 29.86 |
| Esophagus | **57.83** | 53.67 |
| Stomach | **61.89** | 50.18 |
| Duodenum | 45.41 | **46.04** |
| Left Kidney | **72.22** | 63.82 |
| Mean | **60.22** | 56.44 |

At the organ level, for targets with fixed shapes and large volumes, such as the right and left kidneys, the liver, and the spleen, it can be seen that our model performs well. In addition our model performs well for targets with fixed positions, such as the aorta and inferior vena cava. By observing the images we found that our model does not perform well when dealing with smaller scale targets, especially for (left and right) adrenal glands and gallbladder. This is fully explainable because as we set a large spacing, the feature representation would inevitably be weakened of small-scale targets.

### 4.3   Quantitative results on test set

The overall quantitative results on test set are shown in Table 5.

### 4.4   Segmentation efficiency results

For the efficiency of segmentation, our model predicted 50 validation images using about 5 minutes, which we think is a relatively acceptable time. For the majority of validated images, the time used to predict individual results was within 11 seconds (the mean inference time on the validation set of our method is 11.56 seconds), but for images with large scales, our method used up to 45.45 seconds Although we increased the spacing of the input data to make the image array size smaller, we had to sacrifice the patch size to reduce the GPU memory usage (with a mean of 2036.04 MB and a max of 2067 MB), resulting in a larger number of patches, so our final prediction time is similar to the performance of nnUnet.

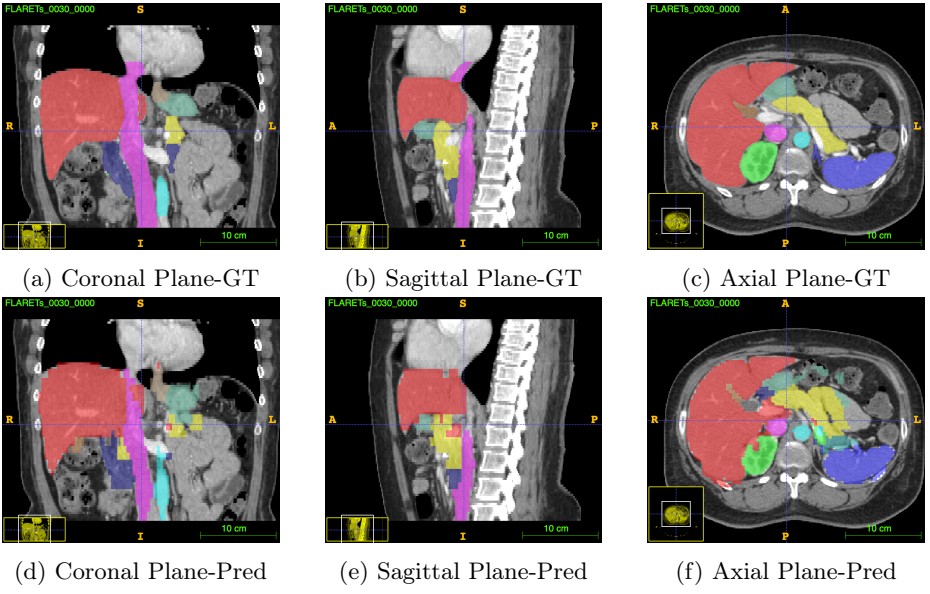

(a) Coronal Plane-GT            (b) Sagittal Plane-GT            (c) Axial Plane-GT

(d) Coronal Plane-Pred          (e) Sagittal Plane-Pred          (f) Axial Plane-Pred

Fig. 2: Standard Validation Case 00030

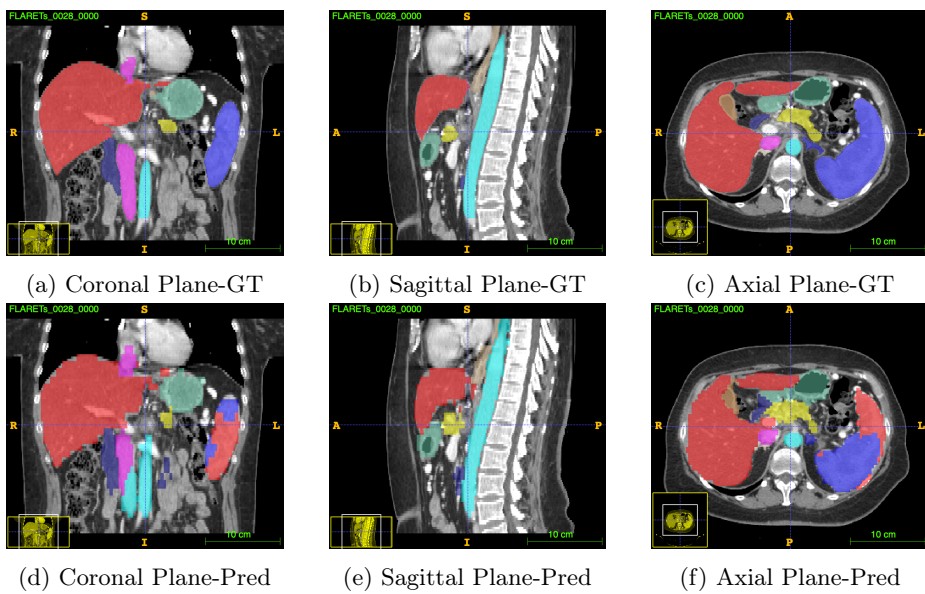

(a) Coronal Plane-GT            (b) Sagittal Plane-GT            (c) Axial Plane-GT

(d) Coronal Plane-Pred          (e) Sagittal Plane-Pred          (f) Axial Plane-Pred

Fig. 3: Standard Validation Case 00028

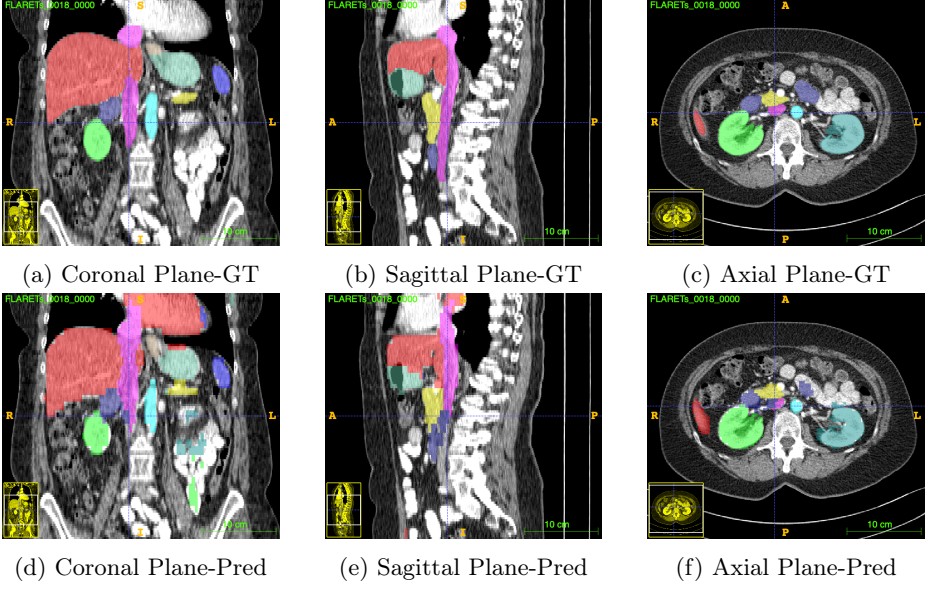

(a) Coronal Plane-GT          (b) Sagittal Plane-GT          (c) Axial Plane-GT

(d) Coronal Plane-Pred          (e) Sagittal Plane-Pred          (f) Axial Plane-Pred

Fig. 4: Bias Validation Case 00018

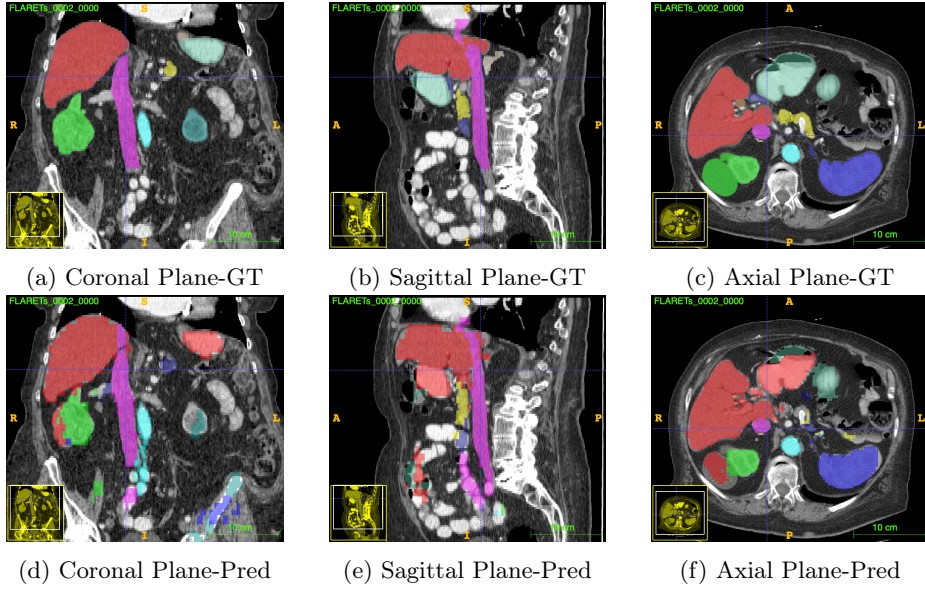

(a) Coronal Plane-GT          (b) Sagittal Plane-GT          (c) Axial Plane-GT

(d) Coronal Plane-Pred          (e) Sagittal Plane-Pred          (f) Axial Plane-Pred

Fig. 5: Bias Validation Case 00002

### 4.5   Limitations and future work

As mentioned in Section 4.2 and Section 4.4, our model had to compromise the spacing after resampling and the size of the patches entering the neural network

Table 5: Quantitative results on test set.

| Organ | DSC(%) | NSD (%) |
|---|---|---|
| Liver | 80.12±10.56 | 68.61±14.39 |
| Right Kidney | 64.59±24.25 | 55.27±24.22 |
| Spleen | 73.44±22.86 | 65.14±22.52 |
| Pancreas | 50.61±16.43 | 63.11±17.45 |
| Aorta | 78.00±14.41 | 74.46±16.35 |
| Inferior Vena Cava | 67.08±15.68 | 59.61±16.41 |
| Right Adrenal Gland | 41.91±15.82 | 56.39±19.22 |
| Left Adrenal Gland | 38.44±19.45 | 51.00±23.74 |
| Gallbladder | 35.14±26.80 | 25.82±19.69 |
| Esophagus | 52.70±15.39 | 64.44±16.50 |
| Stomach | 52.73±19.33 | 48.11±18.38 |
| Duodenum | 41.87±15.75 | 62.04±16.08 |
| Left Kidney | 68.61±14.39 | 60.38±23.44 |
| Mean | 58.14±18.47 | 58.03±19.11 |

in order to improve the computational speed and reduce the computational consumption, which resulted in our model's ability to handle small-scale targets becoming extremely poor.

To solve this problem, our subsequent work has two general directions: one is to reduce the spacing appropriately to find the optimal parameter settings to balance the computational consumption and accuracy (we have tried smaller spacing, which will undoubtedly improve the segmentation accuracy significantly), and the other is to use a cascade model following nnUNet's practice to add an additional neural network structure for small-size targets.

In addition to optimization in terms of network structure, we can also do more experiments in data augmentation methods. At this stage, we have only used conventional and simple data augmentation methods. Due to time constraints, we did not have time to implement more complex enhancement methods such as CutOut or CutMix.

## 5    Conclusion

In conclusion, this work uses the classical Unet model and the Mean Teacher strategy to implement a semi-supervised abdominal organ segmentation task. We do not use complex model structures or difficult-to-deploy usage methods for unlabeled data because we adhere to the idea that for medical images, which usually have relatively fixed structures, good results should be obtained even using simple designs. This idea is also in line with the core idea of the nnUnet model [7], which has been most widely used in recent years. In addition, we slightly sacrifice the accuracy of small target segmentation to obtain a smaller model size and less computational resources.

**Acknowledgements** The authors of this paper declare that the segmentation method they implemented for participation in the FLARE 2022 challenge has not used any pre-trained models nor additional datasets other than those provided by the organizers.

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
