# OpenReview forum: "A Simple Mean-Teacher UNet Model for Efficient Abdominal Organ Segmentation"
_MICCAI.org/2022/Challenge/FLARE_

### Official Review · Reviewer_4eP6 · 2022-09-14
**The authors propose a semi-supervised scheme based on the Mean Teacher model, where a 2D U-Net architecture is trained to segment the abdominal organs.**

**Rating:** 7
**Confidence:** 3

**Review:**

Strengths: The unlabelled data is well used. The model description is easy to follow. The qualitative analysis is well done.
Weaknesses: Some minor grammatical mistakes. Include better descriptions in the figure titles.
Details:
- Specify what type of noise is added to the input images to train the teacher model.
- Include an image of the segmentation network utilized (UNet)
- Defines the loss functions
- Add a better explanation in Fig 1. title.

---

> ### Author Response · Authors · 2022-10-14
> **Re: The authors propose a semi-supervised scheme based on the Mean Teacher model, where a 2D U-Net architecture is trained to segment the abdominal organs.**
>
> For weakness:
> 1. Yes, we forgot to add this description, it was merely an additive Gaussian white noise.
> 2. Since it was only a classical UNet, we felt that it was not necessary to repeat this structure.
> 3. Loss function is also relatively simple, only a classical Dice-Loss, we forgot to mention that, too.
> 4. We will add a brief description to it, thanks
>
> Thanks a lot for your revision!

---

### Official Review · Reviewer_R6hS · 2022-09-16
**A Simple Mean-Teacher UNet Model for Efficient Abdominal Organ Segmentation**

**Rating:** 4
**Confidence:** 3

**Review:**

Strengths:
  The proposed method uses a simple scheme that consists of a classic network (UNet) and semi-supervised learning (Mean-Teacher), and it achieves efficient inference with GPU < 2GB and mean inference time ~10 s.

Weaknesses:
  The segmentation results are not effective enough (0.5893 DSC) relative to the efficiency of the proposed method, probably mainly due to too sparse resampling. Resampling to a spacing of (4, 4, 10) may be too much of a sacrifice, especially on the z-axis. Also, the paper does not contain any ablation study or comparison with other methods, including a baseline fully supervised method.

---

> ### Author Response · Authors · 2022-10-14
> **Re: A Simple Mean-Teacher UNet Model for Efficient Abdominal Organ Segmentation**
>
> For weakness:
> 1. Since we sacrificed the details of the model to achieve this inference speed by minimizing the network, so the accuracy would be lower.
> 2. We do not apply other comparisons with other methods, only compared with/without unlabeled data. In fact without the unlabeled data is exactly a fully supervised method.
>
> Thanks for your revision.

---

### Official Review · Reviewer_hcUj · 2022-09-16
**A Simple Mean-Teacher UNet Model for Efficient Abdominal Organ Segmentation**

**Rating:** 3
**Confidence:** 3

**Review:**

Strengths: The authors design a classical UNet model using the Mean-Teacher strategy to obtain relatively satisfactory segmentation (58.93% DSC and 59.54% NSD) results on a semi-supervised abdominal segmentation dataset.
Weaknesses:
- The overall network architecture is presented in Fig. 1, but the structures of student model and teacher model are not clear.
- The input of teacher model is added noise, but there is no description about noise design.
- The second column in Table 4 is not consistent with the results in Table 3. The ‘with labeled data’ should be ‘with unlabeled data’.

---

> ### Author Response · Authors · 2022-10-14
> **Re: A Simple Mean-Teacher UNet Model for Efficient Abdominal Organ Segmentation**
>
> For weakness:
> 1. The structure of both student/teacher models are just classical UNet, as we mentioned in the paper, so we do not wanted to repeat this structure again in the paper.
> 2. The noise is a additive Gaussian white noise, no specific configure included
> 3. Yes, it was a typo, should be "with/without unlabeled data"
>
> Thanks for your revision

---

### Official Review · Reviewer_v8VF · 2022-09-16
**Good work and really fast for inference.**

**Rating:** 8
**Confidence:** 3

**Review:**

In this work, they adopted the mean teacher UNet model for segmentation and achieving  0.589 mean DSC which is good, their inference time is about 10s per case which is really fast!

It will be better if the final mean DSC can be improved up to 0.8！

---

> ### Author Response · Authors · 2022-10-14
> **Re: Good work and really fast for inference**
>
> Since we sacrificed the details of the model to achieve this inference speed by minimizing the network, so the accuracy would be lower.
>
> Thanks a lot for your revision.

---

### Official Review · Reviewer_rZSQ · 2022-09-19
**Average DSC is relatively low, may consider improving by using two-stages training**

**Rating:** 8
**Confidence:** 3

**Review:**

Pros: 1. Unlabeled data is well used by weak-supervised training which has a ~4% boost in DSC.

Cons: 1. Average DSC is relatively low, may consider improving by using two-stages training.

---

> ### Author Response · Authors · 2022-10-14
> **Re: Average DSC is relatively low, may consider improving by using two-stages training**
>
> Since we sacrificed the details of the model to achieve this inference speed by minimizing the network, so the accuracy would be lower.
>
> Thanks a lot for your revision.

---

### Official Review · Reviewer_Q72D · 2022-09-19
**Nice approach to make use of unlabeled data in an online learning fashion, but missing some small details.**

**Rating:** 7
**Confidence:** 4

**Review:**

This paper combines a mean teacher approach with nnU-Net. Following the mean teacher methodology, two identical U-Nets make up the teacher and student, where the teacher's weights are updated using an exponential moving average of the student's parameters. The authors make use of an increasing alpha parameter schedule to make use of increasingly better segmentations of the student. In addition to the dice loss, the authors use a MSE-based consistency loss.

Pros:
- Well-written and clear description of the methodology, esp. Figure 1 makes the training process very clear.
- Thorough evaluation, including detailed discussion of failure cases.

Missing details:
- Exact alpha parameter schedule used (minor).
- In training protocols (table 2), what does 8*3 samples per image refer to? Should this say samples per batch? (minor)
- Details on the network architecture used are missing.
- More details on strategies to arrive at the reported inference times would be great.

Cons:
- DSC performance somewhat worse than standard nnU-Net. Is this due to more efficient / small architectures used?

In summary a nice and well-written paper, would appreciate adding the missing details though, esp. details on architecture and inference strategies.

---

> ### Author Response · Authors · 2022-10-14
> **Re: Nice approach to make use of unlabeled data in an online learning fashion, but missing some small details.**
>
> For Missing details:
> 1. As we only using experiments to achieve the final params, it's kind of random
> 2. 8*(3 samples per image) means that we extract 3 samples from each image and a batch consists of 8 groups of samples
> 3. Since we did not use extra network other than classical UNet so we did not manage to repeat the structure of UNet
> 4. In fact we do not have other strategies to achieve this inference time, we merely use a minor network
>
> For Cons:
> Yes, since we sacrificed the details of the model to achieve this inference speed by minimizing the network, so the accuracy would be lower.
>
> Thanks a lot for your revision

---

### Meta-Review · Program_Chairs · 2022-09-28

**Recommendation:** Major Revision
**Confidence:** 5

**Metareview:**

Reviewers raise many concerns and suggestions. Please address all comments in the revised manuscript.